# Online Hyperparameter Meta-Learning with Hypergradient Distillation

**Hae Beom Lee**[1]**, Hayeon Lee**[1]**, Jaewoong Shin**[3]**,**
**Eunho Yang**[1,2]**, Timothy M. Hospedales**[4,5]**, Sung Ju Hwang**[1,2]

KAIST[1], AITRICS[2], Lunit[3], South Korea,
University of Edinburgh[4], Samsung AI Centre, Cambridge[5], United Kingdom
{haebeom.lee, hayeon926, shinjw148, eunhoy, sjhwang82}@kaist.ac.kr
{t.hospedales}@ed.ac.uk

## Abstract

Many gradient-based meta-learning methods assume a set of parameters that do not participate in inner-optimization, which can be considered as *hyperparameters*. Although such hyperparameters can be optimized using the existing gradient-based hyperparameter optimization (HO) methods, they suffer from the following issues. Unrolled differentiation methods do not scale well to high-dimensional hyperparameters or horizon length , Implicit Function Theorem (IFT) based methods are restrictive for online optimization, and short horizon approximations suffer from short horizon bias. In this work, we propose a novel HO method that can overcome these limitations, by approximating the second-order term with knowledge distillation. Specifically, we parameterize a single Jacobian-vector product (JVP) for each HO step and minimize the distance from the true second-order term. Our method allows online optimization and also is scalable to the hyperparameter dimension and the horizon length. We demonstrate the effectiveness of our method on two different meta-learning methods and three benchmark datasets.

## 1 Introduction

Meta-learning (Schmidhuber, 1987; Thrun & Pratt, 1998) aims to learn a learning process itself over a task distribution. Many gradient-based meta-learning approaches assume a set of parameters that do not participate in inner-optimization (Lee & Choi, 2018; Flennerhag et al., 2019; Raghu et al., 2019) which can be seen as *hyperparameters*. Those hyperparameters are important in helping the inner-learner converge faster and generalize better. As they are usually very high-dimensional such as element-wise learning rates (Li et al., 2017), we cannot meta-learn them with simple hyperparameter optimization (HO) techniques such as random search (Bergstra & Bengio, 2012) or Bayesian optimization (Snoek et al., 2012) due to the too extensive search space.

In this case, we can use gradient-based HO methods that can directly optimize the high-dimensional hyperparameters by minimizing the validation loss w.r.t. the hyperparameters (Bengio, 2000). Due to the expensive computational cost of evaluating the hypergradients (i.e. the gradient w.r.t. the hyperparameters), there has been a lot of efforts to improve the effectiveness and the efficiency of the algorithms.

Table 1: Comparison between the various gradient-based HO algorithms. 1-step denotes one-step lookahead approximation (Luketina et al., 2016).

|  | FMD | RMD | DrMAD | IFT | 1-step | **Ours** |
|---|---|---|---|---|---|---|
| High-dim. | X | O | O | O | O | **O** |
| Online opt. | O | X | X | △ | O | **O** |
| Constant memory | O | X | O | O | O | **O** |
| Horizon $> 1$ | O | O | O | O | X | **O** |

However, unfortunately, none of the existing algorithms satisfy the following criteria at the same time that should be met for their practical use: *1) scalable to hyperparameter dimension, 2) online optimization, 3) memory-efficient, 4) avoid short-horizon bias*. Please See Table 1 for the comparison of existing gradient-based HO algorithms in the above four criteria.

Forward-Mode Differentiation (FMD) (Franceschi et al., 2017) in Table 1 is an algorithm that forward-propagates Jacobians (i.e. derivatives of the update function) from the first to the last step, which is analogous to real-time recurrent learning (RTRL) (Williams & Zipser, 1989) in recurrent neural networks. FMD allows online optimization (i.e. update hyperparameters every inner-step)

with the intermediate Jacobians and also computes the hypergradients over the entire horizon. However, a critical limitation is that the time and space complexity linearly increases w.r.t. the hyperparameter dimension. Thus, we cannot use FMD for solving many practical meta-learning problems that come with millions of hyperparameters, which is the main problem we tackle in this paper.

Secondly, Reverse-Mode Differentiation (RMD) (Maclaurin et al., 2015) back-propagates the Jacobian-vector products (JVPs) from the last to the initial step, which is structurally identical to backprop through time (BPTT) (Werbos, 1990). RMD is scalable to the hyperparameter dimension, but the space complexity linearly increases w.r.t. the horizon length (i.e., the number of inner-gradient steps used to compute the hypergradient). It is possible to reduce the memory burden by checkpointing some of the previous weights and further interpolating between the weights to approximate the trajectory (Fu et al., 2016). However, RMD and its variants are not scalable for online optimization. This is because they do not retain the intermediate Jacobians unlike FMD and thus need to recompute the whole second-order term for every online HO step.

Thirdly, algorithms based on Implicit Function Theorem (IFT) are applicable to high-dimensional HO (Bengio, 2000; Pedregosa, 2016). Under the assumption that the main model parameters have arrived at convergence, the best-response Jacobian, i.e. how the converged model parameters change w.r.t. the hyperparameters, can be expressed by only the information available at the last step, such as the inverse of Hessian at convergence. Thus, we do not have to explicitly unroll the previous update steps. Due to the heavy cost of computing inverse-Hessian-vector product, Lorraine et al. (2020) propose to approximate it by an iterative method, which works well for high-dimensional HO problems. However, still it is not straightforward to use the method for online optimization because of the convergence assumption. That is, computing hypergradients before convergence does not guarantee the quality of the hypergradients.

To our knowledge, the short horizon approximation such as one-step lookahead (1-step in Table 1) (Luketina et al., 2016) is the only existing method that *fully* supports online optimization, while being scalable to the hyperparameter dimension at the same time. It computes hypergradients only over a single update step and ignores the past learning trajectory, which is computationally efficient as only a single JVP is computed per each online HO step. However, this approximation suffers from the short horizon bias (Wu et al., 2018) by definition.

In this paper, we propose a novel HO algorithm that can simultaneously satisfy all the aforementioned criteria for practical HO. The key idea is to distill the entire second-order term into a single JVP. As a result, we only need to compute the single JVP for each online HO step, and at the same time the distilled JVP can consider longer horizons than short horizon approximations such as one-step lookahead or first-order method. We summarize the contribution of this paper as follows:

- We propose *HyperDistill*, a novel HO algorithm that satisfies the aforementioned four criteria for practical HO, each of which is crucial for a HO algorithm to be applied to the current meta-learning frameworks.

- We show how to efficiently distill the hypergradient second-order term into a single JVP.

- We empirically demonstrate that our algorithm converges faster and provides better generalization performance at convergence, with three recent meta-learning models and on two benchmark image datasets.

## 2 RELATED WORK

**Hyperparameter optimization** When the hyperparameter dimension is small (e.g. less than 100), random search (Bergstra & Bengio, 2012) or Bayesian optimization (Snoek et al., 2012) works well. However, when the hyperparameter is high-dimensional, gradient-based HO is often preferred since random or Bayesian search could become infeasible. One of the most well known methods for gradient-based HO are based on Implicit Function Theorem which compute or approximate the inverse Hessian only at convergence. Bengio (2000) computes the exact inverse Hessian, and Luketina et al. (2016) approximate the inverse Hessian with the identity matrix, which is identical to the one-step lookahead approximation. Pedregosa (2016) approximates the inverse Hessian with conjugate gradients (CG) method. Lorraine et al. (2020) propose Neumann approximation, which is numerically more stable than CG approximation. On the other hand, Domke (2012) proposes unrolled differentiation for solving bi-level optimization, and Shaban et al. (2019) analyzes the truncated unrolled differentiation, which is computationally more efficient. Unrolled diffrentiation can be fur-

ther categorized into forward (FMD) and reverse mode (RMD) (Franceschi et al., 2017). FMD is more suitable for optimizing low-dimensional hyperparamters (Im et al., 2021; Micaelli & Storkey, 2020), but RMD is more scalable to the hyperparameter dimension. Maclaurin et al. (2015) proposes a more memory-efficient RMD, which reverses the SGD trajectory with momentum. Fu et al. (2016) further reduce memory burden of RMD by approximating the learning trajectory with linear interpolation. Luketina et al. (2016) can also be understood as a short horizon approximation of RMD for online optimization. Our method also supports online optimization, but the critical difference is that our algorithm can alleviate the short horizon bias (Wu et al., 2018). RMD is basically a type of backpropagation and it is available in deep learning libraries (Grefenstette et al., 2019).

**Meta-learning** Meta-learning (Schmidhuber, 1987; Thrun & Pratt, 1998) aims to learn a model that generalizes over a distribution of tasks (Vinyals et al., 2016; Ravi & Larochelle, 2016). While there exists a variety of approaches, in this paper we focus on gradient-based meta-learning (Finn et al., 2017), especially the methods with high-dimensional hyperparameters that do not participate in inner-optimization. For instance, there have been many attempts to precondition the inner-gradients for faster inner-optimization, either by *warping* the parameter space with every pair of consecutive layers interleaved with a warp layer (Lee & Choi, 2018; Flennerhag et al., 2019) or directly modulating the inner-gradients with diagonal (Li et al., 2017) or block-diagonal matrix (Park & Oliva, 2019). Perturbation function is another form of hyperparameters that help the inner-learner generalize better (Lee et al., 2019; Ryu et al., 2020; Tseng et al., 2020). It is also possible to let the whole feature extractor be hyperparameters and only adapt the last fully-connected layer (Raghu et al., 2019). On the other hand, some of the meta-learning literatures do not assume a task distribution, but tune their hyperparameters with a holdouot validation set, similarly to the conventional HO setting. In this case, the one-step lookahead method (Luketina et al., 2016) is mostly used for scalable online HO, in context of domain generalization (Li et al., 2018), handling class imbalance (Ren et al., 2018; Shu et al., 2019), gradient-based neural architecture search (Liu et al., 2018), and coefficient of norm-based regularizer (Balaji et al., 2018). Although we mainly focus on meta-learning setting in this work, whose goal is to transfer knowledge through a task distribution, it is straightforward to apply our method to conventional HO problems.

## 3 BACKGROUND

In this section, we first introduce RMD and its approximations for efficient computation. We then introduce our novel algorithm that supports high-dimensional online HO over the entire horizon.

### 3.1 HYPERPARAMETER UNROLLED DIFFERENTIATION

We first introduce notations. Throughout this paper, we will specifiy $w$ as *weight* and $\lambda$ as *hyperparameter*. The series of weights $w_0, w_1, w_2 \ldots, w_T$ evolve with the update function $w_t = \Phi(w_{t-1}, \lambda; D_t)$ over steps $t = 1, \ldots, T$. The function $\Phi$ takes the previous weight $w_{t-1}$ and the hyperparameter $\lambda$ as inputs and its form depends on the current mini-batch $D_t$. Note that $w_1, w_2, \ldots, w_T$ are functions w.r.t. the hyperparameter $\lambda$. The question is how to find a good hyperparameter $\lambda$ that yields a good *response* $w_T$ at the last step. In gradient-based HO, we find the optimal $\lambda$ by minimizing the validation loss $\mathcal{L}^{\text{val}}$ as a function of $\lambda$.

$$\min_{\lambda} \mathcal{L}^{\text{val}}(w_T(\lambda), \lambda) \tag{1}$$

Note that we let the loss function $\mathcal{L}^{\text{val}}(\cdot, \cdot)$ itself be modulated by $\lambda$ for generality. According to the chain rule, the hypergradient is decomposed into

$$\frac{d\mathcal{L}^{\text{val}}(w_T, \lambda)}{d\lambda} = \underbrace{\frac{\partial \mathcal{L}^{\text{val}}(w_T, \lambda)}{\partial \lambda}}_{g_T^{\text{FO}}: \text{First-order term}} + \underbrace{\frac{\partial \mathcal{L}^{\text{val}}(w_T, \lambda)}{\partial w_T} \frac{dw_T}{d\lambda}}_{g_T^{\text{SO}}: \text{Second-order term}} \tag{2}$$

On the right hand side, the first-order (FO) term $g_T^{\text{FO}}$ directly computes the gradient w.r.t $\lambda$ by fixing $w_T$. The second-order (SO) term $g_T^{\text{SO}}$ computes the indirect effect of $\lambda$ through the *response* $w_T$. $\alpha_T = \frac{\partial \mathcal{L}^{\text{val}}(w_T, \lambda)}{\partial w_T}$ can be easily computed similarly to $g_T^{\text{FO}}$, but the *response Jacobian* $\frac{dw_T}{d\lambda}$ is more computationally challenging as it is unrolled into the following form.

$$\frac{dw_T}{d\lambda} = \sum_{t=1}^{T} \left( \prod_{s=t+1}^{T} A_s \right) B_t, \quad \text{where} \quad A_s = \frac{\partial \Phi(w_{s-1}, \lambda; D_s)}{\partial w_{s-1}}, \quad B_t = \frac{\partial \Phi(w_{t-1}, \lambda; D_t)}{\partial \lambda} \tag{3}$$

| **Algorithm 1** Reverse-HG (RMD) | **Algorithm 2** DrMAD (Fu et al., 2016) |
|---|---|
| 1: **Input:** The last weight $w_T$ and all the previous weights $w_0, \ldots, w_{T-1}$. | 1: **Input:** The last weight $w_T$ and the initial weight $w_0$. |
| 2: **Output:** Hypergradient $g^{\text{FO}} + g^{\text{SO}}$. | 2: **Output:** Approximated hypergradient $g^{\text{FO}} + g^{\text{SO}}$. |
| 3: $\alpha \leftarrow \frac{\partial \mathcal{L}(w_T, \lambda)}{\partial w_T}$, $g^{\text{FO}} \leftarrow \frac{\partial \mathcal{L}(w_T, \lambda)}{\partial \lambda}$, $g^{\text{SO}} \leftarrow 0$ | 3: $\alpha \leftarrow \frac{\partial \mathcal{L}(w_T, \lambda)}{\partial w_T}$, $g^{\text{FO}} \leftarrow \frac{\partial \mathcal{L}(w_T, \lambda)}{\partial \lambda}$, $g^{\text{SO}} \leftarrow 0$ |
| 4: **for** $t = T$ **downto** 1 **do** | 4: **for** $t = T$ **downto** 1 **do** |
| 5: $\quad g^{\text{SO}} \leftarrow g^{\text{SO}} + \alpha B_t$ | 5: $\quad \hat{w}_{t-1} \leftarrow \left(1 - \frac{t-1}{T}\right) w_0 + \frac{t-1}{T} w_T$ |
| 6: $\quad \alpha \leftarrow \alpha A_t$ | 6: $\quad g^{\text{SO}} \leftarrow g^{\text{SO}} + \alpha \hat{B}_t$ |
| 7: **end for** | 7: $\quad \alpha \leftarrow \alpha \hat{A}_t$ |
| 8: **return** $g^{\text{FO}} + g^{\text{SO}}$ | 8: **end for** |
| | 9: **return** $g^{\text{FO}} + g^{\text{SO}}$ |

Eq. (3) involves the Jacobians $\{A\}$ and $\{B\}$ at the intermediate steps. Evaluating them or their vector products are computationally expensive in terms of either time (FMD) or space (FMD, RMD) (Franceschi et al., 2017). Therefore, how to approximate Eq. (3) is the key to developing an efficient and effective HO algorithm.

## 3.2 Reverse-mode differentiation and its approximations

Basically, RMD is structurally analogous to backpropagation through time (BPTT) (Werbos, 1990). In RMD, we first obtain $\alpha_T = \frac{\partial \mathcal{L}^{\text{val}}(w_T, \lambda)}{\partial w_T}$ and back-propagate $A$ and $B$ from the last to the first step in the form of JVPs (See Algorithm 1). Whereas RMD is much faster than FMD as we only need to compute one or two JVPs per each step, it usually requires to store all the previous weights $w_0, \ldots, w_{T-1}$ to compute the previous-step JVPs, unless we consider reversible training with momentum optimizer (Maclaurin et al., 2015). Therefore, when $w$ is high-dimensional, RMD is only applicable to short-horizon problems such as few-shot learning (e.g. $T = 5$ in Finn et al. (2017)).

**Trajectory approximation.** Instead of storing all the previous weights for computing $A$ and $B$, we can approximate the learning trajectory by linearly interpolating between the last weight $w_T$ and the initial weight $w_0$. Algorithm 2 illustrates the procedure called DrMAD (Fu et al., 2016), where each intermediate weight $w_t$ is approximated by $\hat{w}_t = \left(1 - \frac{t}{T}\right) w_0 + \frac{t}{T} w_T$ for $t = 1, \ldots, T - 1$. $A$ and $B$ are also approximated by $\hat{A}_s = \frac{\partial \Phi(\hat{w}_{s-1}, \lambda; D_s)}{\partial \hat{w}_{s-1}}$ and $\hat{B}_t = \frac{\partial \Phi(\hat{w}_{t-1}, \lambda; D_t)}{\partial \lambda}$, respectively. However, although DrMAD dramatically lower the space complexity, it does not reduce the number of JVPs per each hypergradient step. For each online HO step $t = 1, \ldots, T$ we need to compute $2t - 1$ JVPs, thus the number of total JVPs to complete a single trajectory accumulates up to $\sum_{t=1}^{T}(2t - 1) = T^2$, which is definitely not scalable as an online optimization algorithm.

**Short-horizon approximations.** One-step lookahead approximation (Luketina et al., 2016) is currently one of the most popular high-dimensional online HO method that can avoid computing the excessive number of JVPs (Li et al., 2018; Ren et al., 2018; Shu et al., 2019; Liu et al., 2018; Balaji et al., 2018). The idea is very simple; for each online HO step we only care about the last previous step and ignore the rest of the learning trajectory for computational efficiency. Specifically, for each step $t = 1, \ldots, T$ we compute the hypergradient by viewing $w_{t-1}$ as constant, which yields $\frac{dw_t}{d\lambda} \approx \frac{\partial w_t}{\partial \lambda}\big|_{w_{t-1}} = B_t$ (See Eq. (3)). Or, we may completely ignore all the second-order derivatives for computational efficiency, such that $\frac{dw_t}{d\lambda} \approx 0$ (Flennerhag et al., 2019; Ryu et al., 2020). While those approximations enable online HO with low cost, they are intrinsically vulnerable to short-horizon bias (Wu et al., 2018) by definition.

## 4 Approach

We next introduce our novel online HO method based on knowledge distillation. Our method can overcome all the aforementioned limitations at the same time.

### 4.1 Hypergradient distillation

The key idea is to distill the whole second-order term $g_t^{\text{SO}} = \alpha_t \sum_{i=1}^{t} \left(\prod_{j=i+1}^{t} A_j\right) B_i$ in Eq. (2) into a single JVP evaluated at a distilled weight point $w$ and with a distilled dataset $D$. We denote the normalized JVP as $f_t(w, D) := \sigma(\alpha_t \frac{\partial \Phi(w, \lambda; D)}{\partial \lambda})$ with $\sigma(x) = \frac{x}{\|x\|_2}$. Specifically, we want to solve the following knowledge distillation problem for each online HO step $t = 1, \ldots, T$:

$$\pi_t^*, w_t^*, D_t^* = \arg\min_{\pi, w, D} \left\| \pi f_t(w, D) - g_t^{\text{SO}} \right\|_2 \tag{4}$$

so that we use $\pi_t^* f_t(w_t^*, D_t^*)$ instead of $g_t^{\text{SO}}$. Online optimization is now feasible because for each online HO step $t = 1, \ldots, T$ we only need to compute the single JVP $f_t(w_t^*, D_t^*)$ rather than computing $2t - 1$ JVPs for RMD or DrMAD. Also, unlike short horizon approximations, the whole trajectory information is distilled into the JVP, alleviating the short horizon bias (Wu et al., 2018).

Notice that solving Eq. (4) only w.r.t. $\pi$ is simply a vector projection.

$$\tilde{\pi}_t(w, D) = f_t(w, D)^\top g_t^{\text{SO}}. \tag{5}$$

Then, plugging Eq. (5) into $\pi$ in Eq. (4) and making use of $\|f_t(w, D)\|_2 = 1$, we can easily convert the optimization problem Eq. (4) into the following equivalent problem (See Appendix A).

$$w_t^*, D_t^* = \arg\max_{w, D} \tilde{\pi}_t(w, D), \quad \pi_t^* = \tilde{\pi}_t(w_t^*, D_t^*). \tag{6}$$

$w_t^*$ and $D_t^*$ match the hypergradient *direction* and $\pi_t^*$ matches the *size*.

**Technical challenge.**    However, solving Eq. (6) requires to evaluate $g_t^{\text{SO}}$ for $t = 1, \ldots, T$, which is tricky as $g_t^{\text{SO}}$ is the target we aim to approximate. We next show how to roughly solve Eq. (6) even without evaluting $g_t^{\text{SO}}$ (for $w_t^*, D_t^*$) or by sparsely evaluating an approximation of $g_t^{\text{SO}}$ (for $\pi_t^*$).

## 4.2   Distilling the hypergradient direction

**Hessian approximation.**    In order to circumvent the technical difficulty, we start from making the optimization objective $\tilde{\pi}_t(w, D) = f_t(w, D)^\top g_t^{\text{SO}}$ in Eq. (5) simpler. We approximate $g_t^{\text{SO}}$ as

$$g_t^{\text{SO}} = \alpha_t \sum_{i=1}^{t} \left( \prod_{j=i+1}^{t} A_j \right) B_i \approx \sum_{i=1}^{t} \gamma^{t-i} \alpha_t B_i. \tag{7}$$

with $\gamma \geq 0$, which we tune on a meta-validation set. Note that Eq. (7) is yet too expensive to use for online optimization as it consists of $t$ JVPs. We thus need further distillation, which we will explain later. Eq. (7) is simply a Hessian identity approximation. For instance, vanilla SGD with learning rate $\eta^{\text{Inner}}$ corresponds to $A_j = \nabla_{w_{j-1}}(w_{j-1} - \eta^{\text{Inner}} \nabla \mathcal{L}^{\text{train}}(w_{j-1}, \lambda))$. Approximating the Hessian as $\nabla^2_{w_{j-1}} \mathcal{L}^{\text{train}}(w_{j-1}, \lambda) \approx kI$, we have $A_j = I - \eta^{\text{Inner}} \cdot kI \approx (1 - \eta^{\text{Inner}}k)I = \gamma I$. Plugging Eq. (7) to Eq. (5) and letting $f_t(w_{i-1}, D_i) := \sigma(\alpha_t \frac{\partial \Phi(w_{i-1}, \lambda; D_i)}{\partial \lambda}) = \sigma(\alpha_t B_i)$, we have

$$\tilde{\pi}_t(w, D) \approx \hat{\pi}_t(w, D) = \sum_{i=1}^{t} \delta_{t,i} \cdot f_t(w, D)^\top f_t(w_{i-1}, D_i) \tag{8}$$

where $\delta_{t,i} = \gamma^{t-i} \|\alpha_t B_i\|_2 \geq 0$. Instead of maximizing $\tilde{\pi}_t$ directly, we now maximize $\hat{\pi}_t$ w.r.t. $w$ and $D$ as a proxy objective.

**Lipschitz continuity assumption.**    Now we are ready to see how to distill the hypergradient direction $w_t^*$ and $D_t^*$ *without* evaluating $g_t^{\text{SO}}$. The important observation is that the maximum of $\hat{\pi}_t$ in Eq. (8) is achieved when $f_t(w, D)$ is well-aligned to the other $f_t(w_0, D_1), \ldots, f_t(w_{t-1}, D_t)$. This intuition is directly related to the following Lipschitz continuity assumption on $f_t$.

$$\|f_t(w, D) - f_t(w_{i-1}, D_i)\|_2 \leq K \|(w, D) - (w_{i-1}, D_i)\|_{\mathcal{X}}, \quad \text{for} \quad i = 1, \ldots, t. \tag{9}$$

where $K \geq 0$ is the Lipschitz constant. Eq. (9) captures which $(w, D)$ can minimize $\|f_t(w, D) - f_t(w_{i-1}, D_i)\|_2$ over $i = 1, \ldots, t$, which is equivalent to maximizing $f_t(w, D)^\top f_t(w_{i-1}, D_i)$ since $\|f(\cdot, \cdot)\|_2 = 1$.    For the metric $\|\cdot\|_{\mathcal{X}}$, we let $K^2 \|(w, D)\|_{\mathcal{X}}^2 = K_1^2 \|w\|_2^2 + K_2^2 \|D\|_2^2$ where $K_1, K_2 \geq 0$ are additional constants that we introduce for notational convenience. Taking square of the both sides of Eq. (9) and summing over all $i = 1, \ldots, t$, we can easily derive the following lower bound of $\hat{\pi}_t$ (See Appendix B).

$$2 \sum_{i=1}^{t} \delta_{t,i} - K_1^2 \sum_{i=1}^{t} \delta_{t,i} \|w - w_{i-1}\|_2^2 - K_2^2 \sum_{i=1}^{t} \delta_{t,i} \|D - D_i\|_2^2 \leq \hat{\pi}_t(w, D) \tag{10}$$

We now maximize this lower bound instead of directly maximizing $\hat{\pi}_t$. Interestingly, it corresponds to the following simple minimization problems for $w$ and $D$.

$$\min_{w} \sum_{i=1}^{t} \delta_{t,i} \|w - w_{i-1}\|_2^2, \qquad \min_{D} \sum_{i=1}^{t} \delta_{t,i} \|D - D_i\|_2^2. \tag{11}$$

| **Algorithm 3** HyperDistill | **Algorithm 4** `LinearEstimation`$(\gamma, \lambda, \phi)$ |
|---|---|
| 1: **Input:** $\gamma \in [0, 1]$, initial $\lambda$, and initial $\phi$. | 1: **Input:** $w_0 \leftarrow \phi$ |
| 2: **Output:** Learned hyperparameter $\lambda$. | 2: **for** $t = 1$ **to** $T$ **do** |
| 3: **for** $m = 1$ **to** $M$ **do** | 3: $\quad w_t \leftarrow \Phi(w_{t-1}, \lambda; D_t)$ |
| 4: $\quad$ **if** $m \in$ `EstimationPeriod` **then** | 4: **end for** |
| 5: $\quad\quad \theta \leftarrow$ `LinearEstimation`$(\gamma, \lambda, \phi)$ | 5: $\alpha, \alpha_T \leftarrow \frac{\partial \mathcal{L}^{\text{val}}(w_T, \lambda)}{\partial w_T}, \quad g^{\text{SO}} \leftarrow 0$ |
| 6: $\quad$ **end if** | 6: **for** $t = T$ **downto** $1$ **do** |
| 7: $\quad w_0 \leftarrow \phi$ | 7: $\quad \hat{w}_{t-1} \leftarrow \left(1 - \frac{t-1}{T}\right) w_0 + \frac{t-1}{T} w_T$ |
| 8: $\quad$ **for** $t = 1$ **to** $T$ **do** | 8: $\quad g^{\text{SO}} \leftarrow g^{\text{SO}} + \alpha \hat{B}_t$ (Eq. (16)) |
| 9: $\quad\quad w_t^*, D_t^* \leftarrow$ Eq. (13), Eq. (14). | 9: $\quad \alpha \leftarrow \alpha \hat{A}_t$ |
| 10: $\quad\quad \pi_t^* \leftarrow c_\gamma(t; \theta)$ in Eq. (15) | 10: $\quad s \leftarrow T - t + 1$ |
| 11: $\quad\quad w_t \leftarrow \Phi(w_{t-1}, \lambda; D_t)$ | 11: $\quad w_s^*, D_s^* \leftarrow$ Eq. (17), Eq. (18) |
| 12: $\quad\quad g \leftarrow g_t^{\text{FO}} + \pi_t^* f_t(w_t^*, D_t^*)$ | 12: $\quad v_s \leftarrow \alpha_T \frac{\partial \Phi(w_s^*, \lambda; D_s^*)}{\partial \lambda}$ |
| 13: $\quad\quad \lambda \leftarrow \lambda - \eta^{\text{Hyper}} g$ | 13: $\quad x_s \leftarrow \|v_s\|_2 \cdot \frac{1 - \gamma^s}{1 - \gamma}, \quad y_s \leftarrow \sigma(v_s)^{\mathsf{T}} g^{\text{SO}}$ |
| 14: $\quad$ **end for** | 14: **end for** |
| 15: $\quad \phi \leftarrow \phi - \eta^{\text{Reptile}}(\phi - w_T)$ | 15: **return** $(x^{\mathsf{T}} y)/(x^{\mathsf{T}} x)$ |
| 16: **end for** | |

**Efficient sequential update.** Eq. (11) tells how to determine the distilled $w_t^*$ and $D_t^*$ for each HO step $t = 1, \ldots, T$. Since $\|\alpha_t B_i\|_2$ is expensive to compute, we approximate as $\|\alpha_t B_0\|_2 \approx \|\alpha_t B_1\|_2 \approx \cdots \approx \|\alpha_t B_t\|_2$, yielding the following weighted average as the approximated solution for $w_t^*$.

$$w_t^* \approx \frac{\gamma^{t-1}}{\sum_{i=1}^t \gamma^{t-i}} w_0 + \frac{\gamma^{t-2}}{\sum_{i=1}^t \gamma^{t-i}} w_1 + \cdots + \frac{\gamma^0}{\sum_{i=1}^t \gamma^{t-i}} w_{t-1} \tag{12}$$

The following sequential update allows to efficiently evaluate Eq. (12) for each HO step. Denoting $p_t = (\gamma - \gamma^t)/(1 - \gamma^t) \in [0, 1)$, we have

$$t = 1 : w_1^* \leftarrow w_0, \qquad t \geq 2 : w_t^* \leftarrow p_t w_{t-1}^* + (1 - p_t) w_{t-1} \tag{13}$$

Note that the online update in Eq. (13) does not require to evaluate $g_t^{\text{SO}}$. It only requires to incorporate the past learning trajectory $w_0, w_1, \ldots, w_{t-1}$ through the sequential updates. Therefore, the only additional cost is the memory for storing and updating the weighted running average $w_t^*$.

For $D$, we have assumed Euclidean distance metric as with $w$, but it is not straightforward to think of Euclidean distance between datasets. Instead, we simply interpret $p_t$ and $1 - p_t$ as probabilities with which we proportionally subsample each dataset.

$$t = 1 : D_1^* \leftarrow D_1, \qquad t \geq 2 : D_t^* \leftarrow \text{SS}\left(D_{t-1}^*, p_t\right) \cup \text{SS}\left(D_t, 1 - p_t\right) \tag{14}$$

where $\text{SS}(D, p)$ denotes random `SubSampling` of `round_off`$(|D|p)$ instances from $D$. There may be a better distance metric for datasets and a corresponding solution, but we leave it as a future work. See Algorithm 3 for the overall description of our algorithm, which we name as *HyperDistill*.

**Role of $\gamma$** Note that Eq. (12) tells us the role of $\gamma$ as a *decaying factor*. The larger the $\gamma$, the longer the past learning trajectory we consider. In this sense, our method is a generalization of the one-step lookahead approximation, i.e. $\gamma = 0$, which yields $w_t^* = w_{t-1}$ and $D_t^* = D_t$, ignoring the whole information about the past learning trajectory except the last step. $\gamma = 0$ may be too pessimistic for most of the cases, so we need to find better performing $\gamma$ for each task carefully.

### 4.3 Distilling the hypergradient size

Now we need to plug the distilled $w_t^*$ and $D_t^*$ into $\tilde{\pi}_t(w, D)$ in Eq. (5) to obtain the scaling factor $\pi_t^* = f_t(w_t^*, D_t^*)^{\mathsf{T}} g_t^{\text{SO}}$, for online HO steps $t = 1, \ldots, T$. However, whereas evaluating the single JVP $f_t(w_t^*, D_t^*)$ is tolerable, again, evaluating $g_t^{\text{SO}}$ is misleading as it is the target we aim to approximate. Also, it is not straightforward for $\pi_t^*$ to apply a similar trick we used in Sec. 4.2.

**Linear estimator.** We thus introduce a linear function $c_\gamma(t; \theta)$ that estimates $\pi_t^*$ by periodically fitting $\theta \in \mathbb{R}$, the parameter of the estimator. Then for each HO step $t$ we could use $c_\gamma(t; \theta)$ instead of fully evaluating $\pi_t^*$. Based on the observation that the form of lower bound in Eq. (10) is *roughly* proportional to $\sum_{i=1}^t \delta_{t,i} = \sum_{i=1}^t \gamma^{t-i} \|\alpha_t B_i\|_2$, we conveniently set $c_\gamma(t; \theta)$ to as follows:

$$c_\gamma(t; \theta) = \theta \cdot \|v_t\|_2 \cdot \sum_{i=1}^t \gamma^{t-i}, \quad \text{where} \quad v_t := \alpha_t \frac{\partial \Phi(w_t^*, \lambda; D_t^*)}{\partial \lambda}. \tag{15}$$

**Collecting samples.** We next see how to collect samples $\{(x_s, y_s)\}_{s=1}^T$ for fitting the parameter $\theta$, where $x_s = \|v_s\|_2 \cdot \frac{1-\gamma^s}{1-\gamma}$ and $y_s = \pi_s^* = f_s(w_s^*, D_s^*)^\mathsf{T} g_s^{\mathrm{SO}} = \sigma(v_s)^\mathsf{T} g_s^{\mathrm{SO}}$. For this, we need to *efficiently* collect:

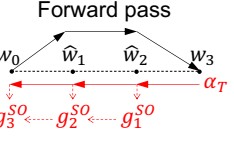

Forward pass

Back-prop. (DrMAD)

Figure 1: Collecting $g_s^{\mathrm{SO}}$

1. $g_s^{\mathrm{SO}}$, the second-order term computed over the horizon of size $s$.
2. $v_s$, the distilled JVP computed over the horizon of size $s$.

**1)** $g_s^{\mathrm{SO}}$: Note that DrMAD in Algorithm 2 (line 6) sequentially back-propagates $g^{\mathrm{SO}}$ for $t = T, \ldots, 1$. The important observation is that, at step $t$, this incomplete second-order term $g^{\mathrm{SO}} = \sum_{i=t}^T \alpha_T \hat{A}_T \hat{A}_{T-1} \cdots \hat{A}_{i+1} \hat{B}_i$ can be seen as the valid second-order term computed over the horizon of size $s = T - t + 1$. This is because the reparameterization $s = T - t + 1$ gives

$$g_s^{\mathrm{SO}} = \sum_{i=1}^s \alpha_{s+(T-s)} \hat{A}_{s+(T-s)} \hat{A}_{s-1+(T-s)} \cdots \hat{A}_{i+1+(T-s)} \hat{B}_{i+(T-s)} \qquad (16)$$

for $s = 1, \ldots, T$, nothing but shifting the trajectory index by $T-s$ steps so that the last step is always $T$. Therefore, we can efficiently obtain the valid second-order term $g_s^{\mathrm{SO}}$ for all $s = 1, \ldots, T$ through the single backward travel along the interpolated trajectory (See Figure 1). Each $g_s^{\mathrm{SO}}$ requires to compute only one or two additional JVPs. Also, as we use DrMAD instead of RMD, we only store $w_0$ such that the memory cost is constant w.r.t. the total horizon size $T$.

**2)** $v_s$: For computing the distilled JVP $v_s$, we first compute the distilled $w_s^*$ and $D_s^*$ as below, similarly to Eq. (13) and (14). Denoting $p_s = (1 - \gamma^{s-1})/(1 - \gamma^s)$, we have

$$s = 1 : w_1^* \leftarrow w_{T-1}, \quad s \geq 2 : w_s^* \leftarrow p_s w_{s-1}^* + (1 - p_s) w_{T-s} \qquad (17)$$

$$s = 1 : D_1^* \leftarrow D_T, \quad s \geq 2 : D_s^* \leftarrow \mathtt{SS}\left(D_{s-1}^*, p_s\right) \cup \mathtt{SS}\left(D_{T-s+1}, 1 - p_s\right) \qquad (18)$$

We then compute the unnormalized distilled JVP as $v_s = \alpha_{s+(T-s)} \frac{\partial \Phi(w_s^*, \lambda; D_s^*)}{\partial \lambda}$.

**Estimating $\theta$.** Now we are ready to estimate $\theta$. For $x$, we have $x_s = \|v_s\|_2 \cdot \frac{1-\gamma^s}{1-\gamma}$ and collect $x = (x_1, \ldots, x_T)$. For $y$, we have $y_s = \sigma(v_s)^\mathsf{T} g_s^{\mathrm{SO}}$ and collect $y = (y_1, \ldots, y_T)$. Finally, we estimate $\theta = (x^\mathsf{T} y)/(x^\mathsf{T} x)$. See Algorithm 3 and Algorithm 4 for the details. Practically, we set $\mathtt{EstimationPeriod}$ in Algorithm 3 to every 50 completions of the inner-optimizations, i.e. $\{1, 51, 101, \ldots\}$. Thus, the computational cost of $\mathtt{LinearEstimation}$ is marginal in terms of the wall-clock time (see Table 3).

## 5 EXPERIMENTS

**Baselines.** We demonstrate the efficacy of our algorithm by comparing to the following baselines.
**1) First-Order Approximation (FO).** Computationally the most efficient HO algorithm that completely ignores the second-order term, i.e. $g^{\mathrm{SO}} = 0$. **2) One-step Look-ahead Approximation (1-step).** (Luketina et al., 2016) The short-horizon approximation where only a single step is unrolled to compute each hypergradient. **3) DrMAD.** (Fu et al., 2016) An approximation of RMD that linearly interpolates between the initial and the last weight to save memory (see Algorithm 2).
**4) Neumann IFT (N.IFT).** (Lorraine et al., 2020) An IFT based method that approximates the inverse-Hessian-vector product by Neumann series. Note that this method supports online optimization around convergence. Specifically, among total $T = 100$ inner-steps, N.IFT$(N, K)$ means for the last $K$ steps we perform online HO each with $N$ inversion steps. It requires total $(N + 1) \times K$ JVPs. We tune $(N, K)$ among $\{(2, 25), (5, 10), (10, 5)\}$, roughly computing 50 JVPs per inner-opt.
**5) HyperDistill.** Our high-dimensional online HO algorithm based on the idea of knowledge distillation. We tune the decaying factor within $\gamma \in \{0.9, 0.99, 0.999, 0.9999\}$. The linear regression is done every 50 inner-optimization problems.

**Target meta-learning models.** We test on the following three meta-learning models.
**1) Almost No Inner Loop (ANIL).** (Raghu et al., 2019) The intuition of ANIL is that the need for task-specific adaptation diminishes when the task distribution is homogeneous. Following this intuition, based on a typical 4-layer convolutional network with 32 channels (Finn et al., 2017), we designate the three bottom layers as the high-dimensional hyperparameter and the 4th convolutional layer and the last fully connected layer as the weight, similarly to Javed & White (2019).

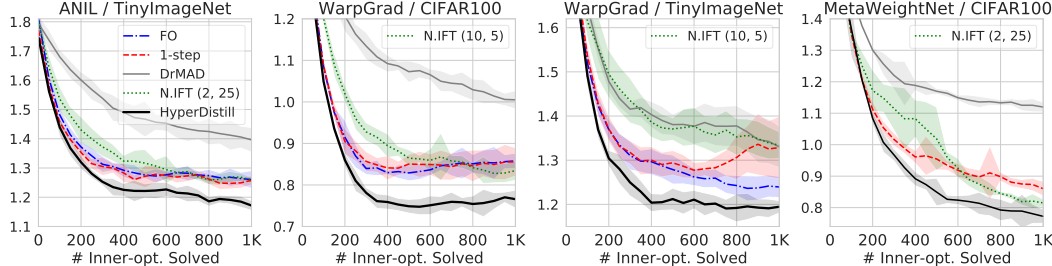

Figure 2: **Meta-training convergence** measured in $\mathcal{L}^{\text{val}}(w_T, \lambda)$ with $T = 100$ inner-steps. We report mean and and 95% confidence intervals over 5 meta-training runs.

| | Online optim. | # JVPs / inner-opt. | ANIL tinyImageNet | WarpGrad CIFAR100 | WarpGrad tinyImageNet | MetaWeightNet CIFAR100 |
|---|---|---|---|---|---|---|
| FO | O | 0 | $53.62_{\pm 0.06}$ | $58.16_{\pm 0.52}$ | $53.54_{\pm 0.74}$ | N/A |
| 1-step | O | 50 | $53.90_{\pm 0.43}$ | $58.18_{\pm 0.52}$ | $49.97_{\pm 2.46}$ | $58.45_{\pm 0.40}$ |
| DrMAD | X | 199 | $49.84_{\pm 1.35}$ | $55.13_{\pm 0.64}$ | $50.71_{\pm 1.16}$ | $57.03_{\pm 0.42}$ |
| Neumann IFT | △ | $\{55, 60, 75\}$ | $53.76_{\pm 0.31}$ | $58.88_{\pm 0.65}$ | $50.15_{\pm 0.98}$ | $59.34_{\pm 0.27}$ |
| **HyperDistill** | O | $\approx 58$ | $\mathbf{56.37_{\pm 0.27}}$ | $\mathbf{60.91_{\pm 0.27}}$ | $\mathbf{55.04_{\pm 0.52}}$ | $\mathbf{60.82_{\pm 0.33}}$ |

Table 2: **Meta-test performance** measured in test classification accuracy (%). We report mean and and 95% confidence intervals over 5 meta-training runs.

**2) WarpGrad.** (Flennerhag et al., 2019) Secondly, we consider WarpGrad, whose goal is to meta-learn non-linear warp layers that facilitate fast inner-optimization and better generalization. We use 3-layer convolutional network with 32 channels. Every layer is interleaved with two warp layers that do not participate in the inner-optimization, which is the high-dimensional hyperparameter.

**3) MetaWeightNet.** (Shu et al., 2019) Lastly, we consider solving the label corruption problem with MetaWeightNet, which meta-learns a small MLP taking a 1D loss as an input and output a reweighted loss. The parameter of the MLP is considered as a high-dimensional hyperparameter. Labels are independently corrupted to random classes with probability 0.4. Note that we aim to meta-learn the MLP over a task distribution and apply to diverse unseen tasks, instead of solving a single task. Also, in this meta model the direct gradient is zero, $g^{\text{FO}} = 0$. In this case, $\pi^*$ in HyperDistill has a meaning of nothing but rescaling the learning rate, so we simply set $\pi^* = 1$.

**Use of Reptile.** Note that for all the above meta-learning models, we meta-learn the weight initialization with Reptile (Nichol et al., 2018) as well, representing a more practical meta-learning scenario than learning from random initialization. We use the Reptile learning rate $\eta^{\text{Reptile}} = 1$. Note that $\phi$ in Algorithm 3 and Algorithm 4 denotes the Reptile initialization parameter.

**Task distribution.** We consider the following two datasets. To generate each task, we randomly sample 10 classes from each dataset (5000 examples) and randomly split them into 2500 training and 2500 test examples. **1) TinyImageNet.** (Le & Yang, 2015) This dataset contains 200 classes of general categories. We split them into 100, 40, and 60 classes for meta-training, meta-validation, and meta-test. Each class has 500 examples of size $64 \times 64$. **2) CIFAR100.** (Krizhevsky et al., 2009) This dataset contains 100 classes of general categories. We split them into 50, 20, and 30 classes for meta-training, meta-validation, and meta-test. Each class has 500 examples of size $32 \times 32$.

**Experimental setup.** **Meta-training:** For **inner-optimization** of the weights, we use SGD with momentum 0.9 and set the learning rate $\mu^{\text{Inner}} = 0.1$ for MetaWeightNet and $\mu^{\text{Inner}} = 0.01$ for the others. The number of inner-steps is $T = 100$ and batchsize is 100. We use random cropping and horizontal flipping as data augmentations. For the **hyperparameter optimization**, we also use SGD with momentum 0.9 with learning rate $\mu^{\text{Hyper}} = 0.01$ for MetaWeightNet and $\mu^{\text{Hyper}} = 0.001$ for the others, which we linearly decay toward 0 over total $M = 1000$ inner-optimizations. We perform parallel meta-learning with meta-batchsize set to 4. **Meta-testing:** We solve 500 tasks to measure average performance, with exactly the same inner-optimization setup as meta-training. We repeat this over 5 different meta-training runs and report mean and 95% confidence intervals (see Table 2). **Code is publicly available at:** https://github.com/haebeom-lee/hyperdistill

## 5.1 ANALYSIS

We perform the following analysis together with the WarpGrad model and CIFAR100 dataset.

**HyperDistill provides faster convergence and better generalization.** Figure 2 shows that HyperDistill shows much faster meta-training convergence than the baselines for all the meta-learning

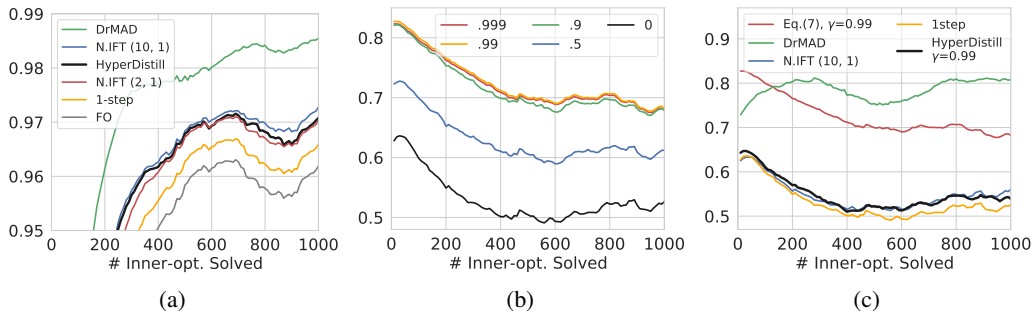

Figure 3: **Cosine similarity to exact RMD** in terms of **(a)** hypergradients $g^{FO} + g^{SO}$. **(b, c)** second-order term $g^{SO}$. The curves in **(b)** correspond to Eq. (7) with various $\gamma$.

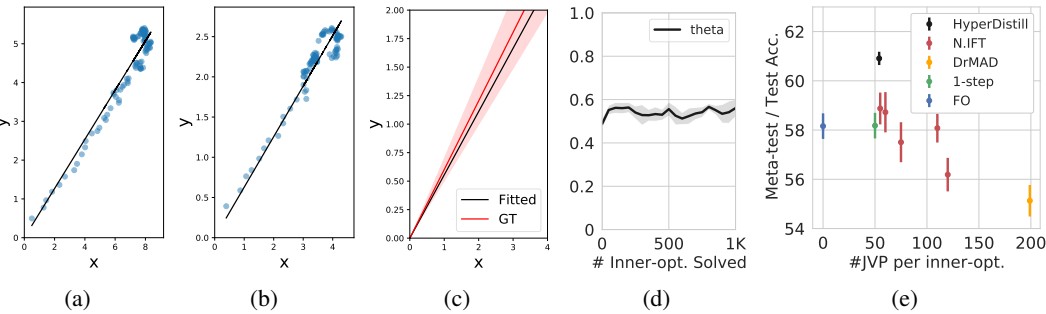

Figure 4: **(a,b)** Samples collected from Algorithm 4 and correspondingly fitted linear estimators ($\theta$). **(c)** A fitted estimator and the range of actual ground-truth estimator (the shaded area is one sigma). **(d)** The stability of $\theta$ estimation. **(e)** Meta-test performance vs. computational cost in terms of the number of JVPs per inner-opt.

models and datasets we considered. We see that the convergence of offline method such as DrMAD is significantly worse than a simple first-order method, demonstrating the importance of frequent update via online optimization. HyperDistill shows significantly better convergence than FO and 1-step because it is online and at the same time alleviates the short horizon bias. As a result, Table 2 shows that the meta-test performance of HyperDistill is significantly better than the baselines, although it requires comparable number of JVPs per each inner-optimiztion.

**HyperDistill is a reasonable approximation of the true hypergradient.** We see from Figure 3(a) that the hypergradient obtained from HyperDistill is more similar to the exact RMD than those obtained from FO and 1-step, demonstrating that HyperDistill can actually alleviate the short horizon bias. HyperDistill is even comparable to N.IFT(10, 1) that computes 11 JVPs, whereas HyperDistill computes only a single JVP. Such results indicate that the approximation we used in Eq. (7) and DrMAD in Eq. (16) are accurate enough. Figure 3(b) shows that with careful tuning of $\gamma$ (e.g. 0.99), the direction of the approximated second-order term in Eq. (7) can be much more accurate than the second-order term of 1-step ($\gamma = 0$). In Figure 3(c), as HyperDistill distills such a good approximation, it can provide a better direction of the second-order term than 1-step. Although the gap may seem marginal, even N.IFT(10, 1) performs similarly, showing that matching the direction of the second-order term without unrolling the full gradient steps is inherently a challenging problem. Figure 4(a) and 4(b) show that the samples collected according to Algorithm 4 is largely linear, supporting our choice of Eq. (15). Figure 4(c) and 4(d) show that the range of fitted $\theta$ is accurate and stable, explaining why we do not have to perform the estimation frequently. Note that DrMAD approximation (Eq. (16)) is accurate (Figure 3(a) and 3(c)), helping to predict the hypergradient size.

**HyperDistill is compuatationally efficient.** Figure 4(e) shows the superior computational efficiency of HyperDistill in terms of the trade-off between meta-test performance and the amount of JVP computations. Note that wall-clock time is roughly proportional to the number of JVPs per inner-optimization. In Appendix F, we can see that the actual increase in memory cost and wall-cock time is very marginal compared to 1-step approximation.

## 6 CONCLUSION

In this work, we proposed a novel HO method, *HyperDistill*, that can optimize high-dimensional hyperparameters in an online manner. It was done by approximating the exact second-order term with knowledge distillation. We demonstrated that HyperDistill provides faster meta-convergence and better generalization performance based on realistic meta-learning methods and datasets. We also verified that it is thanks to the accurate approximations we proposed.

**Acknowledgements** This work was supported by Google AI Focused Research Award, Center for Applied Research in Artificial Intelligence (CARAI) grant funded by DAPA and ADD (UD190031RD), the Engineering Research Center Program through the National Research Foundation of Korea (NRF) funded by the Korean Government MSIT (NRF-2018R1A5A1059921), and Institute of Information & communications Technology Planning & Evaluation (IITP) grant funded by the Korea government(MSIT) (No.2019-0-00075, Artificial Intelligence Graduate School Program(KAIST)).

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

## A    DERIVATION OF EQUATION (6)

Let $f \coloneqq f_t(w, D)$ and $g \coloneqq g_t^{\text{SO}}$ for notational simplicity. Note that $\|f\| = 1$. Then,

$$
\begin{aligned}
\tilde{\pi}_t(w, D) &= \arg\min_{\pi} \|\pi f - g\| \\
&= \arg\min_{\pi} \|\pi f - g\|^2 \\
&= \arg\min_{\pi} \pi^2 f^\mathsf{T} f - 2\pi f^\mathsf{T} g + g^\mathsf{T} g \\
&= f^\mathsf{T} g
\end{aligned}
\tag{19}
$$

Plugging this into $\pi$ in Eq. (19) and with the assumption $\tilde{\pi}(w, D) = f^\mathsf{T} g \geq 0$, we have

$$
\begin{aligned}
w^*, D^* &= \arg\min_{w,D} (f^\mathsf{T} g)^2 \cdot f^\mathsf{T} f - 2(f^\mathsf{T} g) \cdot f^\mathsf{T} g + g^\mathsf{T} g \\
&= \arg\max_{w,D} (f^\mathsf{T} g)^2 \\
&= \arg\max_{w,D} f^\mathsf{T} g \\
&= \arg\max_{w,D} \tilde{\pi}(w, D).
\end{aligned}
\tag{20}
$$

Note that Eq. (20) results from encoding the closed-form solution $\tilde{\pi}_t(w, D)$ already. Therefore, the above is a joint optimization so that we do not have to repeat alternating optimizations between $(w, D)$ and $\pi$.

## B    DERIVATION OF EQUATION (10)

Let $f \coloneqq f_t(w, D)$ and $f_i \coloneqq f_t(w_{i-1}, D_i)$ for notational simplicity. Note that $\|f\| = \|f_1\| = \cdots = \|f_t\| = 1$ and we are given the following $t$ inequalities.

$$
\|f - f_i\| \leq K \|(w, D) - (w_{i-1}, D_i)\|_{\mathcal{X}}, \quad \text{for} \quad i = 1, \dots, t.
$$

Taking square of both sides and multiplying $\delta_{t,i}$,

$$
2\delta_{t,i} - \delta_{t,i} f^\mathsf{T} f_i \leq K_1^2 \delta_{t,i} \|w - w_{i-1}\|^2 + K_2^2 \delta_{t,i} \|D - D_i\|^2, \quad \text{for} \quad i = 1, \dots, t.
$$

Summing the $t$ inequalities over all $i = 1, \dots, t$,

$$
\begin{aligned}
&2\sum_{i=1}^{t} \delta_{t,i} - \sum_{i=1}^{t} \delta_{t,i} f^\mathsf{T} f_i \\
&\leq K_1^2 \sum_{i=1}^{t} \delta_{t,i} \|w - w_{i-1}\|^2 + K_2^2 \sum_{i=1}^{t} \delta_{t,i} \|D - D_i\|^2.
\end{aligned}
$$

Rearranging the terms,

$$
\begin{aligned}
&2\sum_{i=1}^{t} \delta_{t,i} - K_1^2 \sum_{i=1}^{t} \delta_{t,i} \|w - w_{i-1}\|^2 - K_2^2 \sum_{i=1}^{t} \delta_{t,i} \|D - D_i\|^2 \\
&\leq \sum_{i=1}^{t} \delta_{t,i} f^\mathsf{T} f_i \\
&= \hat{\pi}(w, D)
\end{aligned}
$$

## C  META-VALIDATION PERFORMANCE

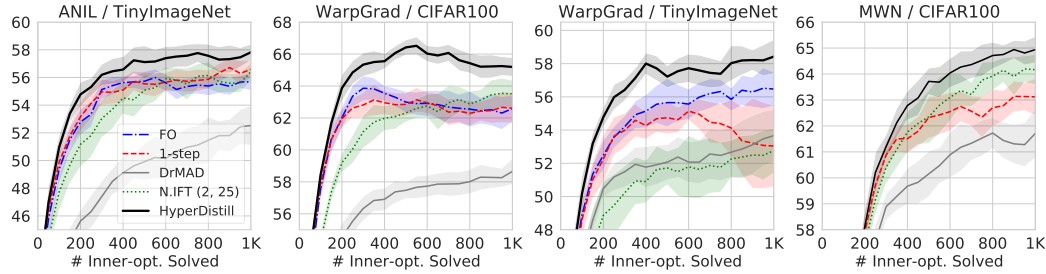

Figure 5: **Meta-validation performance**. We report mean and and 95% confidence intervals over 5 meta-training runs.

Figure 5 shows the meta-validation performance as the meta-training proceeds. We can see that our HyperDistill shows much faster meta-convergence and shows better generalization at convergence than the baselines, which is consistent with the meta-training convergence shown in Figure 2.

## D  HYPER-HYPERPARAMETER ANALYSIS

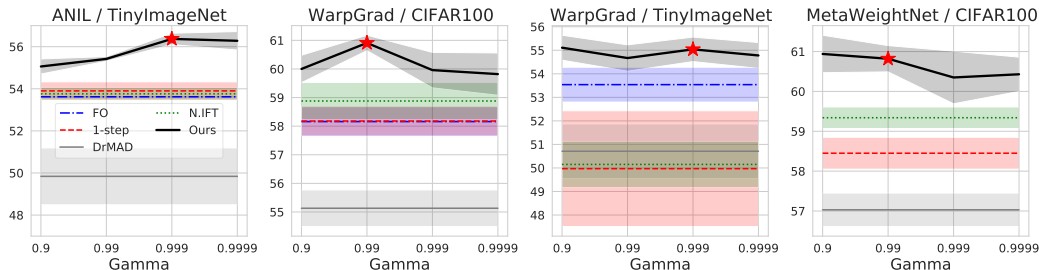

Figure 6: **Meta-test performance** by varying the value of $\gamma$. Red stars denote the actuall $\gamma$ we used for each experiment (we found them with a meta-validation set) and the corresponding performance.

Our algorithm, HyperDistill has a hyper-hyperparamter $\gamma$ that we tune with a meta-validation set in the range $\{0.9, 0.99, 0.999, 0.9999\}$. Figure 6 shows that with all the values of $\gamma$ and for all the experimental setups we consider, HyperDistill outperforms all the baselines with significant margins. This demonstrates that the performance of HyperDistill is not much sensitive to the value of $\gamma$.

## E  MORE DETAILS OF METAWEIGHTNET EXPERIMENTS

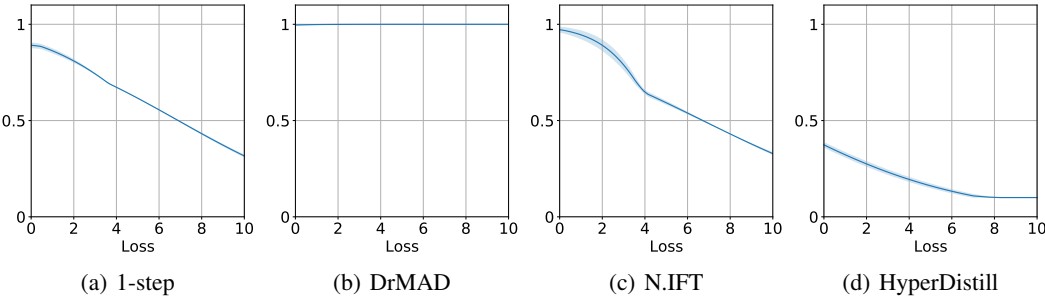

(a) 1-step          (b) DrMAD          (c) N.IFT          (d) HyperDistill

Figure 7: **Learned loss weighting function** with each algorithm.

We provide the additional experimental setup for the MetaWeightNet (Shu et al., 2019) experiments. We use $1 - 200(\text{ReLU}) - 1$ loss weighting network architecture, following the original paper. Also, we found that lower bounding the output of the weighting function with $0.1$ can stabilize the training. Figure 7 shows the resultant loss weighting function learned with each algorithm. We see that the learned weighting function with HyperDistill tend to output lower values than the baselines.

## F    COMPUTATIONAL EFFICIENCY

|  | ANIL | WarpGrad | | MetaWeightNet |
|  | tinyImageNet | CIFAR100 | tinyImageNet | CIFAR100 |
|  | (Mb) / (s / inner-opt.) | (Mb) / (s / inner-opt.) | (Mb) / (s / inner-opt.) | (Mb) / (s / inner-opt.) |
|---|---|---|---|---|
| FO | 1430 / 6.23 | 1092 / 5.24 | 1840 / 7.01 | N/A |
| 1-step | 1584 / 6.80 | 1650 / 6.88 | 3844 / 18.81 | 1214 / 6.17 |
| DrMAD | 1442 / 20.88 | 1734 / 19.83 | 4148 / 57.09 | 1262 / 17.57 |
| Neumann IFT | 1392 / 7.98 | 1578 / 7.43 | 3286 / 21.49 | 1262 / 6.93 |
| **HyperDistill** | 1638 / 6.92 | 1714 / 8.68 | 4098 / 22.15 | 1206 / 6.04 |

Table 3: **Memory and wall-clock time** required by a single process. We used RTX 2080 Ti for the measurements. Note that we run 4 processes in parallel in our actual experiments (meta-batchsize is 4), which requires roughly ×4 memory than the values reported in this table.

Table 3 shows the computational efficiency measured in actual memory usage and average wall-clock time required to complete a single inner-optimization. We can see from the table that whereas our HyperDistill requires slightly more memory and wall-clock time than 1-step or Neumann IFT method, the additional cost is definitely tolerable considering the superior meta-test performance shown in Table 2.

## G    SINUSOIDAL REGRESSION

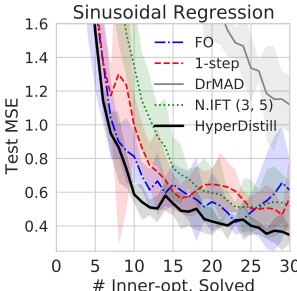

Figure 8: Meta-convergence

|  | MSE |
|---|---|
| FO | $0.567_{\pm 0.193}$ |
| 1-step | $0.670_{\pm 0.283}$ |
| DrMAD | $1.086_{\pm 0.176}$ |
| N.IFT | $0.502_{\pm 0.146}$ |
| **HyperDistill** | $\mathbf{0.327_{\pm 0.052}}$ |

Table 4: Meta-test performance.

In this section, we conduct sinusoidal experiments to demonstrate the efficacy of our method on a regression task.

**Task distribution.**    Each task is to regress a curve sampled from the following distribution of sinusoidal functions; the amplitude and phase is sampled from $\mathcal{U}(0.1, 5)$ and $\mathcal{U}(0, \pi)$, respectively. The range of input is $[-5, 5]$, and the input and output dimensions are both 1 (Finn et al., 2017). We consider 10-shot regression problems.

**Meta-model and network architecture.**    We set the meta-model to ANIL (Raghu et al., 2019). Given the 4-layer fully-connected ReLU network (1-100-100-100-1), the first three layers are set to the hyperparameter, and only the last layer is adapted to given tasks.

**Experimental setup.**    For **inner-optimization** of the weights, we use SGD with momentum 0.9 and set the learning rate to $\mu^{\text{Inner}} = 0.01$ The number of inner-steps is $T = 30$. For the **hyperparameter optimization**, we use Adam optimizer (Kingma & Ba, 2015) with the learning rate $\mu^{\text{Hyper}} = 0.001$. The number of inner-optimizations solved per each meta-convergence is $M = 30$. We perform parallel meta-learning with the meta-batchsize set to 10. **Meta-testing:** We solve 1000 tasks to measure average mean squared error (MSE), with exactly the same inner-optimization setup as meta-training. We repeat this over 5 different meta-training runs and report mean and 95% confidence intervals (see Table 4).

**Results.**    In Figure 8 and Table 4, we see that HyperDistill shows better meta-convergence and meta-test performance than all the baselines. Comparing to FO and 1-step baselines, we see that it is still important to consider longer horizons even in this relatively fewer-shot learning scenario. Also, DrMAD shows poor performance, demonstrating the importance of online optimization.

## H   STANDARD LEARNING SCENARIO

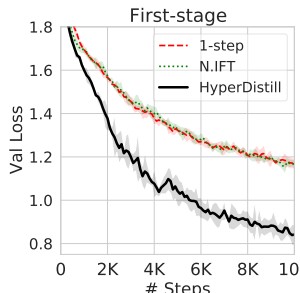
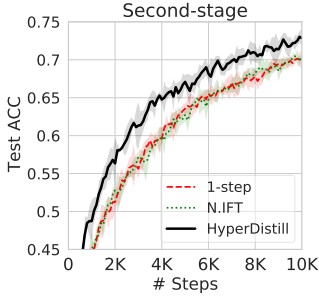

|  | Test ACC |
| --- | --- |
| 1-step | $70.15_{\pm 1.24}$ |
| N.IFT | $70.85_{\pm 0.63}$ |
| **HyperDistill** | **$72.68_{\pm 1.15}$** |

Table 5: Meta-test performance.

Figure 9: Meta-train convergence   Figure 10: Meta-test convergence

In this section, instead of meta-learning setting which involves some task distribution, we consider standard learning scenario where we are given only a single classification task.

**Two-stage learning.**   We consider the following two-stage learning scenario, which is a reasonable way to cast a standard classification task into a meta-learning problem (Liu et al., 2018). In **the first-stage** we split the whole CIFAR10 (Krizhevsky et al., 2009) training dataset into two sets with equal number of instances (each with 25,000 instances). We then use one as a training set to optimize the weight $w$ and the other as a validation set to optimize the hyperparameter $\lambda$. In **the second stage**, we merge the two datasets into the original one and re-train $w$ with it from the random initialization, while the learned hyperparameter $\lambda$ in the first stage is fixed.

**MetaWeightNet (Shu et al., 2019).**   Again, we consider MetaWeightNet which we used in the experimental section 5. We use the same network structure for the loss weighting network and the same label corruption strategy. Note that the original paper uses 1-step strategy.

**Experimental setup.**   In the **first-stage**, for both weight $w$ and hyperparameter $\lambda$, we use SGD with momentum 0.9 and set the learning rate to 0.01 The number of training steps is set to $T = 10,000$. In the **second-stage**, as mentioned above, we reinitialize and train $w$ with the merged dataset, while fixing $\lambda$ obtained from the first stage. We repeat this two-stage process 5 times and report mean and 95% confidence intervals (see Table 5). **Hyper-hyperparameters:** For Neumann IFT, we compute the hypergradients (each with 5 inversion steps) for every 5 gradient steps. For HyperDistill, we set $\gamma = 0.9$.

**Results.**   In Figure 9, 10 and Table 5, we see that HyperDistill shows much better meta-convergence and meta-test performance than all the baselines. The results demonstrate the effectiveness of our method for solving standard HO problems. Note that we cannot consider FO because there is no direct gradient, i.e. $g^{\text{FO}} = 0$ for this MetaWeightNet model. Also, we do not consider the offline methods like DrMAD because now the horizon length became too long to backpropagate all the way through the learning process.

## I   FURTHER ANALYSIS ON SHORT HORIZON BIAS

In this section, we demonstrate the effect of short horizon bias by showing the convergence plots with the varying decaying factor $\gamma$ (see Figure 11). Note that in this controlled experiment we want to see the effect of $\gamma$ only, so we fix the scaling factor as $\pi = 1$. In the right Figure 11, $\gamma = 0$ corresponds to 1-step which computes the hypergradients by unrolling only a single step, thus suffers from short horizon bias. As we increase $\gamma$, the convergence roughly improves as well, demonstrating that the short horizon bias can be alleviated by increasing $\gamma$. Also, see Figure 3(b) which shows how well the different values of $\gamma$ can recover the true hypergradients. The best performing $\gamma$ seems 0.99 in terms of the cosine similarity to the true hypergradients.

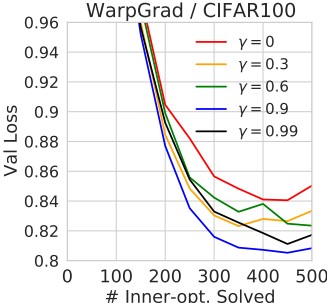

Figure 11: short horizon bias

# J   EXPERIMENTS WITH FOMAML

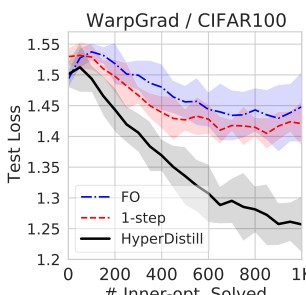

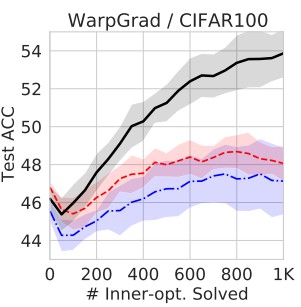

|  | Test ACC |
|---|---|
| FO | $45.01_{\pm 1.29}$ |
| 1-step | $46.03_{\pm 0.59}$ |
| **HyperDistill** | **$51.43_{\pm 0.98}$** |

Table 6: Meta-test performance.

Figure 12: Meta-train convergence   Figure 13: Meta-val. convergence

In order to demonstrate that our HyperDistill works well with other meta-learning algorithms than Reptile (Nichol et al., 2018), we consider first-order MAML (FOMAML) (Finn et al., 2017). Note that we need first-order approximation of MAML because the original MAML with second order derivative is too expensive with the long horizon $T = 100$. We can see from Figure 12 and Figure 13 that our method provides much faster and better meta-convergence than FO and 1-step. As a result, in Table 6 our HyperDistill achieves significantly better meta-test performance than the baselines. Note that we do not report the performance of DrMAD and Neumann IFT becuase they fail to meta-converge with FOMAML.

Note that the performance with FOMAML is much worse than with Reptile, which is well known results from the previous literature (Flennerhag et al., 2018; Shin et al., 2021). This is because FOMAML ignores the whole learning process except the very last step's gradient information. This becomes more critical with longer horizons as the last step gradient becomes arbitrary uninformative to the initialization. We thus recommend using Reptile instead of FOMAML.

