# OpenReview forum: "Online Hyperparameter Meta-Learning with Hypergradient Distillation"
_ICLR.cc/2022/Conference — ICLR 2022 Spotlight_

### Official Review · Reviewer_GYz2 · 2021-11-02

**Correctness:** 3
**Technical Novelty And Significance:** 3
**Empirical Novelty And Significance:** 3
**Recommendation:** 6
**Confidence:** 3

**Main Review:**

Pos:
1, The topic is interesting and important which can be impactful in machine learning research community.
2, The idea is reasonable and novel. Applying knowledge distillation in second-order gradient computation is novel.

Cons:
1, The written can be more clear by simplifying the notations.
2, The experiments can be extensive by applying the methods in more datasets, for instance, meta-dataset.

**Summary Of The Paper:**

This paper proposes a new online hyperparameter optimization algorithm. Applying meta-learning for hyperparameter optimization is reasonable and interesting but suffers from the second-order gradient computation. Implicit Function Theorem and Unrolled differentiation can be used to approximate the meta-gradient but also causes various problems. In this paper, the authors propose an interesting method by approximating the second-order approximation with knowledge distillation.

**Summary Of The Review:**

I think the topic is novel and impactful and the idea is novel and interesting. Experiments show the effectiveness of the method. Hence, I recommend acceptance.

---

> ### Author Response · Authors · 2021-11-16
> **Response to Reviewer GYz2**
>
> We sincerely appreciate your suggestions and insightful comments. We could improve our paper based on your feedback. Below we respond to each of your comments.
>
> **[Q1]** The writing can be clearer by simplifying the notations.
> - Thank you for the suggestion. We will try our best to simplify notations as much as possible. However, please also understand that due to the nature of this topic, it is very hard to explain things without introducing many variables and terms. Also please consider that **the other reviewers (w7Q6, zUNV, G8V6) found the paper well presented and clearly explained**.
>
> ---
>
> **[Q2]** The experiments can be extensive by applying to more datasets, e.g. Meta-dataset.
> - Thank you for the suggestion. **Meta-dataset is a few-shot learning benchmark**, which seems a little bit orthogonal to the direction of our paper which aims to solve many-shot and long-horizon problems. Instead, according to the reviewer w7Q6’s suggestion, we further verify our method on the following three additional settings:
>
> - **1) In Appendix G** in the revision, we conduct sinusoidal experiments following Finn et al. [1] to demonstrate the effectiveness of our method on a **regression task**. The Figure 8 and Table 4 show that HyperDistill provides better meta-convergence and meta-test performance than the baselines. The results demonstrate that **our method can be applied to general types of tasks.**
>
>
> |        | MSE    |
> |--------------|:---------:|
> | FO         |     0.567 $\pm$ 0.193 |
> | 1-step      |    0.670 $\pm$ 0.283 |
> | DrMAD      |   1.086 $\pm$ 0.176 |
> | N.IFT      |    0.502 $\pm$ 0.146 |
> | **HyperDistill** | **0.327 $\pm$ 0.052** |
>
> - **2) In Appendix H** in the revision, instead of a meta-learning scenario where we are given a task distribution, we further verify our method on a **standard learning scenario** where we are given only a single classification task. We consider MetaWeightNet to solve the label corruption problem. From Figure 9, 10 and Table 5, we see that HyperDistill shows much better meta-convergence and the meta-test performance than all the baselines. The results demonstrate that **our method can be applied to standard learning scenarios as well as meta-learning settings.**
>
> |        | Test ACC    |
> |--------------|:---------:|
> | FO         |     45.01 $\pm$ 1.29 |
> | 1-step      |    46.03 $\pm$ 0.59 |
> | **HyperDistill** | **51.43 $\pm$ 0.98** |
>
> - **3) In Appendix J** in the revision, we demonstrate that our HyperDistill works well with **FOMAML**, another meta-learning algorithm than Reptile. We can see from Figure 12, Figure 13, and Table 6 that HyperDistill provides much better meta-convergence and meta-test performance than the baselines. The results demonstrate that **our method is compatible with general types of meta-learning algorithms.**
>
>
> |        | Test ACC    |
> |--------------|:---------:|
> | 1-step         |   70.15 $\pm$ 1.24 |
> | N.IFT      |    70.85 $\pm$ 0.63 |
> | **HyperDistill** | **72.68 $\pm$ 1.15** |
>
> # References
> [1] Finn et al., Model-Agnostic Meta-Learning for Fast Adaptation of Deep Networks, ICML 2017.

---

### Official Review · Reviewer_G8V6 · 2021-11-02

**Correctness:** 3
**Technical Novelty And Significance:** 3
**Empirical Novelty And Significance:** 3
**Recommendation:** 8
**Confidence:** 2

**Main Review:**

Strengths
-------------
- This paper is very clearly presented; the background is appropriately detailed and provides sufficient information to compare HyperDistill to the relevant literature. The theory underlying the distillation process is well-explained and easy to follow; the experiments are well-described, easy to interpret, and illustrate well the various advantages of HyperDistill as it compares to other modern methods.

- HyperDistill addresses the shortcomings that are pointed out in the other modern gradient-based HO methods (scalability; limited horizon; applicability to the online setting). In practice, across a variety of datasets, HyperDistill consistently achieves lower validation loss and improves test accuracy.

- HyperDistill is in part motivated by having a long-term horizon (in comparison to the one-step lookahead approach). The authors compare explicitly to this baseline, and show that HyperDistill achieves a better reconstruction of the gradient, and recover one-step lookahead as a special case.

- The set of experiments is Experimentally, HyperDistill improves over the many baselines that are considered by the authors are well-designed, consider a variety of relevant baselines, and presents unambiguously the advantages of HyperDistill.

Weaknesses
-----------------
The main weakness that I see in this work lies in the creation of the distilled dataset D (section 4.2). The authors elide over the difficulty of building this dataset, mentioning only in passing that defining a distance over datasets is difficult. This could be done more carefully, starting by defining precisely what space the minimization problem (11) operates over, and then explicitly solving (or bounding the gap of the proposed solution to) the problem.

Questions / comments
------------------------------
- Nit: it might be worth clarifying the dimensionality and spaces that \pi, \w_t, and \D_t lie in e.g., in (4).

- Similarly, I would clarify the norm being used in (4), especially as the choice of norm becomes important for the distilled dataset later on.

- Some typos: "longer horizonS" in Section 1, "identitcal" in Section 2, "completely ignoreS" in Section 5

**Summary Of The Paper:**

This paper develops a practical gradient-based hyperparameter optimization method, HyperDistill, that meets the following criteria
  a) scalability in hyperparameter dimension and memory constraints,
  b) accuracy (hyper-gradient update terms do not depend on only the last step of gradient updates)
  c) applicability to the online setting.

The main difficulty lies in estimating the gradient of the weights with respect to the hyperparameters. The authors do so by approximating it as a single Jacobian-vector product, using a "distilled" weight and dataset pair.

Experimentally, HyperDistill achieves better validation losses and higher quality true hypergradient estimates than a variety of recent, relevant baselines, and does in an efficient fashion.

**Summary Of The Review:**

The paper is clearly written, the theoretical justification for the paper is well-presented and intuitive, and experiments confirm the value of the proposed method.

My only criticism is the comparative lack of care with which the distilled dataset is analyzed; this stands out particularly in contrast to how the  rest of the technical difficulties are addressed.

---

> ### Author Response · Authors · 2021-11-16
> **Response to Reviewer G8V6**
>
> We sincerely appreciate your suggestions and insightful comments. We could improve our paper based on your feedback. Below we respond to each of your comments.
>
> **[Q1]** The main weakness that I see in this work lies in the creation of the distilled dataset D (section 4.2). The authors elide over the difficulty of building this dataset, mentioning only in passing that defining a distance over datasets is difficult. This could be done more carefully, starting by defining precisely what space the minimization problem (11) operates over, and then explicitly solving (or bounding the gap of the proposed solution to) the problem.
>
> - We appreciate your insightful comments. Let us first clarify the search space of the distilled dataset $D$. In this paper, we want to find a distilled dataset $D \subset D^\text{train}$ such that $\|D\| =\texttt{batchsize}$ and where $D^\text{train} = \\{(x_n,y_n)\\}_{n=1}^{N}$ is the entire training dataset of the given task. Then, for the online optimization steps $t=1,\dots,T$, we should continuously update this distilled subset so that it is as “close” as possible to the past and current data batches $D_1,D_2,\dots,D_t$, each of which is a subset of $D^\text{train}$, i.e. we should solve Eq.(11). However, as we mentioned in the paper, it is not straightforward to define the “closeness” between the two datasets. We need some reference to further proceed the derivation.
>
> - For this reason, we argue that the solution Eq.(14) can be a simple and intuitive ad-hoc solution which allows the distilled dataset to be aware of the relative importance across the batches, $\frac{\gamma^{t-1}}{\sum_{i=1}^t \gamma^{t-i}}, \frac{\gamma^{t-2}}{\sum_{i=1}^t \gamma^{t-i}},\dots,\frac{\gamma^0}{\sum_{i=1}^t \gamma^{t-i}}$. Eq.(14) is equivalent to the proportional subsampling from the candidate datasets $D_1,D_2,\dots,D_t$ with the above ratio, which is analogous to the weighted average solution in Eq.(12).
>
> - However, it is also true that this dataset distillation part lacks rigorous mathematical derivation, so it’s the main limitation of our method. Recently, we found one paper [1] defining the notion of distance between datasets via optimal transport. We are currently investigating if we can use this distance to better formulate the minimization problem (11) and then explicitly solve it as you suggested. This problem of online dataset distillation seems technically challenging, but we will include it in the revision if we succeed.
>
> ---
>
> **[Q2]** Clarify the dimensionality, spaces, and norms. Correct some typos.
> - Thank you for pointing them out. We have corrected them in the revision.
>
> # Reference
> [1] Alverez-Melis and Fusi, Geometric Dataset Distances via Optimal Transport, NeurIPS 2020

---

### Official Review · Reviewer_zUNV · 2021-11-03

**Correctness:** 4
**Technical Novelty And Significance:** 2
**Empirical Novelty And Significance:** 3
**Recommendation:** 8
**Confidence:** 3

**Main Review:**

Pros:
* The paper is well-written and easy to follow. The introduction and related work well summarize the current state of the problem and motivate the proposed method.
* The empirical experiments are rigorous and well supports the claims made in the paper.

Concerns:
* One question I had was on the definition of the short-horizon bias. I believe that the short-horizon bias occurs for a particular type of hyperparameter such as learning rate. For many regularization hyperparameters, my understanding was we still don’t know the greedy update performs worse in the long run. I think a clarification of the definition in the main text would be helpful.
* In Figure 2, what would happen to the convergence if we vary \gamma? As the \gamma gets smaller, does the convergence look more similar to the 1-step or DrMAD? I think this experiment would strengthen the claim that HyperDistill is robust against short-horizon bias.
* What are additional hyperparameters HyperDistill introduce? Does HyperDistill robust to this new set of hyperparameters? Since the theory part of the algorithm mostly relies on several approximations/assumptions, I believe that the ablation study on all datasets would be helpful.

Minor comments/questions:
* In the introduction, it says “computing hypergradients before convergence does not guarantee the quality of the hypergradients”. I was wondering if this statement is true. The IFT works the same way even if the training loss is a constant above 0.
* In the experiment section, under the task distribution paragraph, TinyImageNet is not cited properly.
* In appendix E, it would be helpful to directly cite the original paper.
* In Figure 7, DrMAD stays at 1 constant. Does it mean that it is not learning anything?


**Summary Of The Paper:**

The paper proposes a novel hyperparameter optimization algorithm in meta-learning to overcome previous limitations of only being able to see a longer horizon and not being scalable to high dimensional hyperparameters. In particular, the authors propose to distill the hypergradient second-order term into a one-step Jacobioan-vector product. The authors show that, with their approximated algorithm, it is possible to perform hyperparameter optimization for higher dimensional hyperparameters and longer horizon length even in an online setting. Empirically, the authors show the advantages of the proposed approach in several benchmark datasets.

**Summary Of The Review:**

Overall, I vote for accepting. I believe that the paper is well-written, well-motivated and the claims made in the paper are well justified empirically. Moreover, the content is relevant to the ICLR community.

---

> ### Author Response · Authors · 2021-11-16
> **Response to Reviewer zUNV**
>
> We sincerely appreciate your suggestions and insightful comments. We could improve our paper based on your feedback. Below we respond to each of your comments.
>
> **[Q1]** A clarification of the definition of short horizon bias.
> - Short horizon bias is the bias caused by a relatively shorter inner-trajectory used to compute each meta-gradient than the actual inner-trajectory. As you pointed out, whereas Wu et al. [1] analyze the effect of short horizon bias for learning rate, it has not been very clear if other types of hyperparameters are subject to the same effect. Indeed, one of the goals of this paper is to show that this is the case. Figure 3(b) in our paper shows that as $\gamma$ decreases as 0.9 $\rightarrow$ 0.5 $\rightarrow$ 0, the approximation Eq.(7) becomes inaccurate accordingly. Since $\gamma$ is roughly analogous to the length of inner-trajectory, we can conclude that the WarpGrad hyperparameter is subject to the short horizon bias effect as well. We will make it clear in the revision.
>
> ---
>
> **[Q2]**  As the $\gamma$ gets smaller, does the convergence look more similar to the 1-step or DrMAD?
> - Thank you for suggesting the experiment. As mentioned above, Figure 3(b) already shows that the quality of the meta-gradients gradually worsen as we decrease $\gamma$, where the worst case $\gamma=0$ corresponds to 1-step.
> - In **Appendix I** in the revision, as you suggested, we further demonstrate this short horizon bias effect by showing the convergence plots with the varying decaying factor $\gamma$. Note that in this controlled experiment we want to see the effect of $\gamma$ only, so we fix the scaling factor as $\pi=1$. We can see from Figure 11 that as we increase $\gamma$, the convergence roughly improves as well, demonstrating that the short horizon bias can be alleviated by increasing $\gamma$.
>
> ---
>
>
> **[Q3]**  What additional hyperparameters does HyperDistill introduce?
> - There is **no other hyper-hyperparameter than $\gamma$**. Note that in Appendix D we empirically demonstrate that HyperDistill is robust to the different values of $\gamma$.
>
> ---
>
> **[Q4]**  Is the following statement true?; “computing hypergradients before convergence does not guarantee the quality of the hypergradients”
> - Yes, it is true. That is the main limitation of IFT based methods and why we often need to sufficiently “warm up” the inner-learner before computing a hypergradient [2]. For this reason, in our experiments Neumann IFT performs meta-update only for the last part of the inner-optimization. See Section 4.3 of Lorraine et al. [3] for more discussion on this limitation of IFT.
>
> **[Q5]**  Incorrect citation for TinyImageNet dataset, missing citation in Appendix E
> - Thank you for pointing those out. We have corrected them in the revision.
>
> **[Q6]**   In Figure 7, DrMAD stays at 1 constant. Does it mean that it is not learning anything?
> - Initially, the output of the weighting function is roughly 0.5 because of the sigmoid function. As the learning proceeds, DrMAD algorithm seems to prefer a higher inner-learning rate for some reason, such that it keeps pushing the output of the weighting function toward 1. It means that it learns something, but the results are suboptimal compared to the other online optimization algorithms.
>
> # References
> [1] Wu et al., Understanding Short-Horizon Bias in Stochastic Meta-Optimization, ICLR 2018
>
> [2] Raghu et al., Meta-Learning to Improve Pre-Training, NeurIPS 2021
>
> [3] Lorraine et al., Optimizing Millions of Hyperparameters by Implicit Differentiation, AISTATS 2020

---

> > ### Comment · Reviewer_zUNV · 2021-11-25
> > **Response to Authors**
> >
> > Thank you for the reply. I acknowledge that I read the authors' responses and other reviewers' comments. I will keep my current score and confidence.

---

### Official Review · Reviewer_w7Q6 · 2021-11-03

**Correctness:** 3
**Technical Novelty And Significance:** 3
**Empirical Novelty And Significance:** 3
**Recommendation:** 6
**Confidence:** 4

**Main Review:**

The proposed method is well-motivated, and the paper is clearly written. My major concerns are about the experiments:

1. In this paper, the authors evaluate the performance on two image classification datasets. It might be more convincing if the authors can evaluate the performance on more kinds of tasks (e.g., regression tasks). Besides, can HyperDistill boost the performance under different meta-learning algorithms? More analysis will make the findings more convincing.

2. HyperDistill is a general hyperparameter optimization method. And the authors mention that the proposed method is easy to be applied on traditional hyperparameter optimization (HPO) tasks. Could you evaluate the effectiveness of the proposed method on some supervised learning datasets and compare it with HPO algorithms.

3. Minor: are the tinyImagenet and CIFAR100 tasks few-shot learning tasks?

---- After Rebuttal---

I am happy with the authors' response and decide to keep my score.


**Summary Of The Paper:**

The authors propose a hyperparameter optimization algorithm in meta-learning, where parameters w/o being involved in the inner loop optimization are treated as hyperparameters. The proposed algorithms approximate the second-order hypergradients via knowledge distillation. They further evaluate the effectiveness on tinyImageNet and CIFAR100.


**Summary Of The Review:**

The motivation of the proposed method is clear, but more comprehensive experiments are needed to verify the effectiveness of Hyperdistill.

---

> ### Author Response · Authors · 2021-11-16
> **Response to Reviewer w7Q6**
>
> We sincerely appreciate your suggestions and insightful comments. We could improve our paper based on your feedback. Below we respond to each of your comments.
>
> **[Q1]** Evaluate the performance on more kinds of tasks? (e.g., regression)
> - Thank you for the suggestion. In **Appendix G** in the revision, we conduct **sinusoidal experiments** following Finn et al. [1] to demonstrate the effectiveness of our method on a regression task. Meta model is set to ANIL. The Figure 8 and Table 4 show that HyperDistill provides better meta-convergence and meta-test performance than the baselines. The results demonstrate that **our method can be applied to general types of tasks.**
>
> |        | MSE    |
> |--------------|:---------:|
> | FO         |     0.567 $\pm$ 0.193 |
> | 1-step      |    0.670 $\pm$ 0.283 |
> | DrMAD      |   1.086 $\pm$ 0.176 |
> | N.IFT      |    0.502 $\pm$ 0.146 |
> | **HyperDistill** | **0.327 $\pm$ 0.052** |
>
> ---
> **[Q2]** Can HyperDistill boost the performance under different meta-learning algorithms?
> - As for the meta-learning algorithms, we demonstrate in **Appendix J** in the revision that our HyperDistill works well with **FOMAML**, another meta-learning algorithm than Reptile. We use FOMAML instead of the second order version of MAML because of the too expensive computational cost with the long horizon (100 steps). We experiment with WarpGrad / CIFAR100. We can see from Figure 12, Figure 13, and Table 6 that HyperDistill provides much better meta-convergence and meta-test performance than the baselines. The results demonstrate that **our method is compatible with general types of meta-learning algorithms.**
> -Note that the performance with FOMAML is much worse than with Reptile, which is well known results from the previous literature [2,3]. This is because FOMAML ignores the whole learning process except the very last step's gradient information. This becomes more critical with longer horizons as the last step gradient becomes arbitrarily uninformative to the initialization. We thus recommend using Reptile instead of FOMAML.
> - In this experiment, we do not report the performance of DrMAD and Neumann IFT because they fail to meta-converge when using FOMAML instead of Reptile.
> - **As for the meta-learning models, we already considered in this paper three modern meta-learning models** such as ANIL, WarpGrad, and MetaWeightNet, on which our HyperDistill significantly outperforms all the other gradient-based HO algorithms.
>
> |        | Test ACC    |
> |--------------|:---------:|
> | FO         |     45.01 $\pm$ 1.29 |
> | 1-step      |    46.03 $\pm$ 0.59 |
> | **HyperDistill** | **51.43 $\pm$ 0.98** |
>
> ---
> **[Q3]** Could you evaluate the effectiveness of the proposed method on some supervised learning datasets and compare it with HPO algorithms.
> - Thank you for the suggestion. In **Appendix H**  in the revision, we verify our method on a **standard learning scenario** where we are given only a single classification task. We consider **MetaWeightNet** to solve the label corruption problem. We use CIFAR10 dataset. From Figure 9, 10 and Table 5, we see that HyperDistill shows much better meta-convergence and the meta-test performance than all the baselines. The results demonstrate that **our method can be applied to standard learning scenarios as well as meta-learning settings.**
> - In this experiment, we cannot consider FO because there is no direct gradient, i.e. $g^\text{FO} = 0$ for this MetaWeightNet model. Also, we do not consider the offline methods like DrMAD because now the horizon length became too long to backpropagate all the way through the learning process.
>
> |        | Test ACC    |
> |--------------|:---------:|
> | 1-step         |   70.15 $\pm$ 1.24 |
> | N.IFT      |    70.85 $\pm$ 0.63 |
> | **HyperDistill** | **72.68 $\pm$ 1.15** |
>
> ---
> **[Q4]** Are the tinyImagenet and CIFAR100 tasks few-shot learning tasks?
> - No, they are bigger than the usual few-shot learning tasks. In the “Task distribution” section on page 8, we describe that each task consists of 10 clases, total 2500 / 2500 instances for training / test set, respectively (10-way 250-shot classification). In this **many-shot learning scenario**, we should have a longer horizon (100 steps) to sufficiently adapt, and therefore the short horizon bias issue becomes relevant.
> ---
> # References
> [1] Finn et al., Model-Agnostic Meta-Learning for Fast Adaptation of Deep Networks, ICML 2017.
>
> [2] Flannerhag et al., Transferring Knowledge across Learning Processes, ICLR 2019
>
> [3] Shin et al., Large-scale Meta-Learning with Continual Trajectory Shifting, ICML 2021

---

### Author Response · Authors · 2021-11-17
**Summary of the Revision**

We really appreciate all the reviewers for their constructive comments. We have responded to the individual comments from the reviewers below, and believe that we have successfully responded to most of them. We have included the discussions and results of the suggested experiments in the revision. Here we briefly summarize the updates we have made to the revision:

- **[Appendix G]** Sinusoidal regression experiments (Reviewer w7Q6)

- **[Appendix H]** Experiments with a standard learning scenario (Reviewer w7Q6)

- **[Appendix I]** Further analysis on short horizon bias (Reviewer zUNV)

- **[Appendix J]** Experiments with FOMAML (w7Q6)

---

### Decision · Program_Chairs · 2022-01-20

**Decision:**

Accept (Spotlight)

**Comment:**

This paper presents a novel methodology for performing meta learning for gradient-based hyperparameter optimization.  The approach overcomes limitations (scaling, e.g.) of previous methods through distilling the gradients of the hyperparameters.  The paper received 4 reviews, of which all were positive (6, 6, 8, 8).  The reviewers appreciated the technical clarity of the paper and found the proposed approach sensible, novel, technically sophisticated and effective.  The main concerns were regarding the comprehensiveness of the experiments and technical presentation of the dataset distillation.  It seems that the reviewers found the author response (lots of results were added) satisfactory regarding these points.  Thus the recommendation is to accept.